# Understanding microRNAs in the Context of Infection to Find New Treatments against Human Bacterial Pathogens

**DOI:** 10.3390/antibiotics11030356

**Published:** 2022-03-08

**Authors:** Álvaro Mourenza, Blanca Lorente-Torres, Elena Durante, Jesús Llano-Verdeja, Jesús F. Aparicio, Arsenio Fernández-López, José A. Gil, Luis M. Mateos, Michal Letek

**Affiliations:** 1Departamento de Biología Molecular, Área de Microbiología, Universidad de León, 24071 León, Spain; amouf@unileon.es (Á.M.); bloret00@estudiantes.unileon.es (B.L.-T.); e.durante2@campus.uniurb.it (E.D.); jllanv00@estudiantes.unileon.es (J.L.-V.); jesus.aparicio@unileon.es (J.F.A.); jagils@unileon.es (J.A.G.); 2L’Università di Urbino Carlo Bo, Via Aurelio Saffi, 2, 61029 Urbino, Italy; 3Departamento de Biología Molecular, Área de Biología Celular, Universidad de León, 24071 León, Spain; arsenio.fernandez@unileon.es; 4Instituto de Biomedicina (IBIOMED), Universidad de León, 24071 León, Spain; 5Neural Therapies SL, Campus de Vegazana s/n, 24071 León, Spain; 6Instituto de Biología Molecular, Genómica y Proteómica (INBIOMIC), Universidad de León, 24071 León, Spain; 7Instituto de Desarrollo Ganadero y Sanidad Animal (INDEGSAL), Universidad de León, 24071 León, Spain

**Keywords:** miRNAs, pathogen, bacteria, infection, antimicrobial

## Abstract

The development of RNA-based anti-infectives has gained interest with the successful application of mRNA-based vaccines. Small RNAs are molecules of RNA of <200 nucleotides in length that may control the expression of specific genes. Small RNAs include small interference RNAs (siRNAs), Piwi-interacting RNAs (piRNAs), or microRNAs (miRNAs). Notably, the role of miRNAs on the post-transcriptional regulation of gene expression has been studied in detail in the context of cancer and many other genetic diseases. However, it is also becoming apparent that some human miRNAs possess important antimicrobial roles by silencing host genes essential for the progress of bacterial or viral infections. Therefore, their potential use as novel antimicrobial therapies has gained interest during the last decade. The challenges of the transport and delivery of miRNAs to target cells are important, but recent research with exosomes is overcoming the limitations in RNA-cellular uptake, avoiding their degradation. Therefore, in this review, we have summarised the latest developments in the exosomal delivery of miRNA-based therapies, which may soon be another complementary treatment to pathogen-targeted antibiotics that could help solve the problem caused by multidrug-resistant bacteria.

## 1. Introduction

Small RNAs are non-coding molecules of RNA of less than 200 nucleotides in length and with important roles in transcriptional regulation. There are different small RNAs, such as small interference RNAs (siRNAs), piwi-interacting RNAs (piRNAs), and micro RNAs (miRNAs). MicroRNAs were identified in the early 1990s [1], and their function as transcriptional regulators was gradually elucidated [2]. MiRNAs are typically around 18–25 nucleotides long non-coding molecules that act as transcriptional regulators by targeting specific messenger RNAs (mRNAs) for their destruction to achieve gene silencing [3,4]. MicroRNA biogenesis is an important process that finalizes with the RNA-Induced Silencing Complex (RISC) formation, which localizes and binds miRNA with its target mRNA. This results in the degradation of the targeted mRNA(s) and a subsequent reduction in the expression of the affected gene(s) [5,6,7]. However, the silencing of a transcriptional repressor may result in downstream gene upregulation. Therefore, some miRNAs can also trigger the expression of specific genes [8].

Most mammalian mRNAs possess conserved targets for miRNAs [9]. The perfect match between a miRNA and its target (3’UTR region of the mRNA) results in mRNA cleavage, eventually leading to gene silencing. However, there is also a possibility of a non-perfect match between a miRNA and the 5’UTR region of a gene [10]. The complementarity degree between a miRNA and its mRNA target dictates its level of degradation or silencing [7,11]. In addition, some post-transcriptional alterations could change the processing of miRNAs by the DROSHA/DICER complex and their loading onto Argonaute (AGO) proteins, an essential component of RISC. These changes in miRNA maturation may alter the miRNA-mediated regulation of gene expression and could be different depending on the type of cell or their microenvironment [7].

Despite their highly complex and understudied roles, it is now becoming clear that miRNAs are essential molecules that regulate multiple molecular pathways in humans and other organisms. There are >2500 human miRNAs annotated in public repositories, and >3000 miRNAs have been additionally identified in specific cell types. These >5500 miRNAs have >45,000 gene targets, representing more than 60% of all human protein-coding genes [4,9,12,13,14]. 

Initially, the role of miRNAs on infection was discovered in models of viral infections [15]. Plants and animals express miRNAs that target viral genes to combat infections caused by a wide range of viruses [16,17]. Antiviral miRNAs may also control the levels of mRNAs produced by the infected host cell, and they play major roles in viral pathogenesis [18]. This led to the development of novel antiviral strategies based on miRNAs [19] and even the design of attenuated vaccines based on miRNA technology [20]. 

Interestingly, the first miRNA related to bacterial infections was found in plants, when miR-393 was discovered as a contributor to the resistance to infection caused by *Pseudomonas syringae* in *Arabidopsis thaliana* [21,22]. The discovery of natural antimicrobial responses based on miRNAs opened the door to new studies in the field of immunology focused on the role of miRNAs in the activation of the immune response [5].

MicroRNA expression is tightly controlled in cells and is tissue- or even organ-specific [7]. Factors that regulate a miRNA expression and activity include genetic polymorphisms, DNA methylation, asymmetric miRNA strand selection, and the miRNA interactions with RNA-binding proteins or other RNAs [7].

Based on this preliminary evidence, the roles of miRNAs were perceived as a new opportunity to discover biomarkers and new therapeutic strategies against a wide range of other pathogens, including bacteria and parasites [5,23,24,25,26,27]. The host miRNA response to bacterial infection was initially studied by stimulation of toll-like receptors (TLRs) with pathogen-associated molecular patterns (PAMPs) and subsequent analysis of the expression profile of different miRNAs [5,28]. One of the earliest studies discovered that miR-146 and miR-155 work as a negative-feedback loop to stop the TLR4-mediated cellular response in human monocytes exposed to lipopolysaccharide (LPS) [28]. Later work has focused on studying the miRNAs involved in the intracellular infection of different bacterial pathogens [23]. Here, we have summarized the latest developments on the role of miRNA in bacterial infections. 

## 2. Human miRNAs and Pathogen Infections

In the context of infection, miRNAs may regulate innate immune pathways that control the magnitude of host inflammatory responses by altering different signalling pathways. However, other miRNAs regulate the expression of specific genes that control pathways relevant to the host cell’s infection. Therefore, we have divided the following section into two subsections to shed light on this matter. The first is dedicated to the role of microRNAs on inflammation, and the second is focused on the role of specific microRNAs in the fine-tuning of the infected host cell. In addition, we have created Table 1 to summarize current knowledge about the miRNAs involved in bacterial infections.

### 2.1. Role of miRNAs in the Regulation of Host Inflammatory Responses during Bacterial Infection

#### 2.1.1. *Mycobacterium tuberculosis*

By far, the most studied relationship between miRNAs and bacteria during infection are those related to *Mycobacterium tuberculosis* [15]. Phagosome rupture is a critical event in mycobacterial infections in which miRNAs play an important role. For example, EsxA and EsxB are essential virulence factors of *M. tuberculosis* that play a role as phagosome maturation inhibitors and miRNA regulators. The deletion of the *esxBA* genes results in the upregulation of mir-206, miR-147 and miR-148a, which play essential roles in the release of many inflammatory cytokines [48,49,61]. 

In addition, miR-20a-3p, miR-99b and miR-1178 are overexpressed in *M. tuberculosis*-infected cells and reduce the immune response to facilitate host colonisation. In particular, miR-20a-3p controls the host immunity by blocking the production of pro-inflammatory cytokines through the control of the IKK/NF-kB pathway [43]. At the same time, miR-99b upregulation blocks the expression of pro-inflammatory cytokines via MyD88 signalling [47]. Finally, miR-1178 targets the TLR4 to block the immune response in *M. tuberculosis*-infected cells, increasing the pathogen’s survival rate [52].

However, the roles of some miRNAs in tuberculosis are still unclear, and their expression is host cell-specific [24]. For example, miR-155 facilitates cell survival and bacterial propagation in macrophages, but it promotes cytokine production and bacterial clearance in T cells via different metabolic routes [50]. 

#### 2.1.2. *Francisella tularensis*

Interestingly, miR-155 has also been identified as a baffling immune response regulator during the infection of other pathogens. In particular, miR-155 is upregulated during infections caused by *Francisella tularensis* subspecies *novicida*. This is less virulent than other subspecies of *F. tularensis*, which do not induce the expression of miR-155. This observation suggests that there are virulence factors involved in controlling the expression of miR-155 that are only present in the most virulent *F. turalensis* subspecies to favour the infection [34,62]. The role of miR-155 in the non-virulent *F. tularensis* ssp. *novicida* is related to the downregulation of SHIP in monocytes and macrophages, which eventually enhances the expression of pro-inflammatory cytokines through the activation of the TLR2/MyD88 pathway [62]. In contrast, *F. tularensis* virulent strains show a marked decrease in miR-155 expression with a concomitant reduction of anti-inflammatory cytokines mediated by the silencing of SHIP-1 and MyD88 [34].

#### 2.1.3. *Vibrio cholerae*

miR-155 is also involved in other infections. For example, *Vibrio cholerae* releases outer membrane vesicles (OMV) that carry different virulence factors. These OMVs elicit the expression of miR-155 and miR-146a in host cells. The expression of these miRNAs eventually results in the downregulation of the inflammatory response, which facilitates bacterial proliferation [57].

#### 2.1.4. *Staphylococcus aureus*

miR-155 overexpression can cause fatal pneumonia in *Staphylococcus aureus* infected patients because of the overexpression of different interleukins, resulting in a fatal cytokine storm [63]. *S. aureus* also elicits the expression of miR-127, and its upregulation may increase the natural antibacterial response to the pathogen in mice by means of STAT3 ubiquitination [56].

#### 2.1.5. *Helicobacter pylori*

*Helicobacter pylori* can also control inflammation by the upregulation of miR-155 [5,64]. This, in turn, reduces the expression of MyD88, whose gene silencing lowers the levels of the pro-inflammatory cytokine IL-8 [65]. In addition, miR-155 is also part of negative feedback that finally results in the downregulation of other inflammatory cytokines [66]. 

However, *H. pylori* chronic infection can cause other disorders such as coronary heart disease [67]. This is also mediated by microRNAs, particularly by the activation and the production of exosomal packaged miR-25, which increases the expression of inflammatory factors in vascular endothelial cells [35]. Moreover, miR-21, miR-218 and miR-223 are also overexpressed in gastric cancer patients during an *H. pylori* infection, and they are probably oncogenic [68]. Because of this, these miRNAs are used as biomarkers of *H. pylori*-induced gastric inflammation and gastric cancer. These data shed some light on the complex relationship between bacterial infections and other pathologies linked to miRNA changes. 

#### 2.1.6. *Chlamydia trachomatis*

*Chlamydia trachomatis* is a human intracellular pathogen and the causative agent of trachoma. The infection caused by *C. trachomatis* is related to the differential expression of miRNAs involved in inflammation [69,70]. In particular, the overexpression of miR-155 and downregulation of miR-184 are associated with inflammation in trachoma patients [33]. 

Moreover, the miRNA expression profile could be used to determine the severity of the disease. Specific miRNA expression patterns are good prognostic markers of pelvic inflammatory disease, a sign of severe genital infection [71]. Additionally, miRNAs that control the NF-kB pathway, such as miR-9, miR-19 and miR-451, are also upregulated during infection [32].

#### 2.1.7. Broad-Spectrum miRNAs

Some miRNAs, however, were identified as having a broad spectrum of antimicrobial effects. In particular, miR-30e-5p reduces bacterial survival by targeting SOCS1 and SOCS3 [72], two crucial regulators of innate immunity whose silencing reduces bacterial replication [73]. 

Other miRNAs target general regulators of the immune response during infection, which may be broad-spectrum miRNAs. For example, some miRNAs target signalling pathways activated by TOLL receptors (TLRs) [25]. In particular, miR-124 is often overexpressed during bacterial infections [25,58,59], and it modulates the immune response negatively through the TLRs/NK-κB signalling pathway. Thus, anti-miR-124 could be used as a broad-spectrum therapy, as previously demonstrated with *Mycobacterium bovis* (BCG) [58]. In addition, the expression of miR-302b is induced by TLR2 and the TLR4/ NK-κB pathway during *Pseudomonas aeruginosa* infection, and its overexpression activates cytokine release [60]. Other miRNAs control the expression of interferon -γ [39]. For example, miR-29 is downregulated during infection of *L. monocytogenes* and *M. bovis* [25,39]. Therefore, it is worth exploring if any of these miRNAs could be used against other bacterial pathogens.

### 2.2. Role of miRNAs in the Control of the Infected Host Cell

#### 2.2.1. *Mycobacterium tuberculosis*

Autophagy plays an important role during intracellular infection caused by *M. tuberculosis*, and miRNAs regulate this molecular pathway. The expression of miR-155 and miR-17-5p reduces the intracellular colonisation of *M. tuberculosis* by modulating different metabolic routes that result in autophagy activation in macrophages [24,41,50]. The upregulation of miR-155 results in the activation of autophagy and a concomitant mycobacterial clearance. In particular, miR-155 binds to the 3’UTR region of the *Rheb* gene, promoting phagosome maturation, binding to lysosomes, and subsequent mycobacterial elimination [51]. 

In addition, *M. tuberculosis* can also control the expression of different miRNAs to reduce autophagy. For instance, miR-27 is upregulated during *M. tuberculosis* infection and downregulates calcium-associated autophagy [45]. The target of miR-27 is the Ca^2+^ transported CAC-NA2D3, which is located at the endoplasmic reticulum (ER) and whose downregulation inhibits autophagosome formation [45]. Moreover, *M. tuberculosis* host cell infection induces the expression of miR-1958, which binds to the 3’UTR region of Atg5, whose silencing results in the inhibition of the autophagic flux [53]. Finally, miR-18a facilitates *M. tuberculosis* infection by silencing the ataxia–telangiectasia-mutated (*ATM*) gene, which decreases LC3-II levels in infected cells and stops the xenophagy process [42].

On the other hand, miR-33 is overexpressed during *M. tuberculosis* infection, and it targets different host cell genes involved in cholesterol transport and fatty acid oxidation, including *ABCA1*, *CROT*, *CPT1*, *HADHB* and *PRKAA1*. This, in turn, activates the lipid catabolism in the infected host cells, which facilitates bacterial colonisation because of the highly lipid-dependent metabolism of *M. tuberculosis* [46]. Thus, an anti-miR-33 may promote phagosome maturation and bacterial clearance by stopping the lipid metabolism of the infected host cell.

#### 2.2.2. Adherent–Invasive *Escherichia coli*

Some microRNAs are also relevant in the Adherent–Invasive *Escherichia coli* (AIEC) colonisation of intestinal mucosa in Crohn’s disease patients [74]. In this context, exosomes carrying miR-30c and miR-130a are released into non-infected cells to silence the expression of ATG5 and ATG16L1. The resulting inhibition of the autophagic flux facilitates the intracellular replication of AIEC [29]. 

#### 2.2.3. *Legionella pneumophila*

Interestingly, *Legionella pneumophila* may control the expression of 85 different miRNAs during infection. In particular, the upregulation of three miRNAs (miR-125b, miR-221, and miR-579) in a cooperative manner leads to the downregulation of the RNA receptor DDX58/RIG-I, the tumour suppressor TP53, the antibacterial LGALS8 and the MX dynamin-like GTPase 1 (MX1), which altogether enhance the intracellular replication of the pathogen [36]. The repressive effects of miR-125b and miR-221 on *MX1* and miR-579 on *LGALS8* are particularly significant. These genes form a newly discovered cellular immune response pathway against *L. pneumophila* whose overexpression results in bacterial clearance [36]. 

#### 2.2.4. *Chlamydia trachomatis*

*C. trachomatis* maintains mitochondrial ATP production during infection through the upregulation of miR-30c-5p. This miRNA downregulates p53, which in turn leads to the downregulation of Drp1, a mitochondrial fission regulator [31]. Many other intracellular pathogens have very tight interactions with mitochondria during host cell infection, which opens the way to interventions aimed at disrupting bacterial proliferation [75].

#### 2.2.5. *Shigella flexneri* and *Salmonella enterica* Serovar Typhimurium

High-throughput screenings have quickly identified novel microRNAs involved in other bacterial infections. One such study has uncovered three important miRNAs expressed during *Shigella flexneri* infection: miR-3668, miR-4732-5p and miR-6073. These miRNAs constrain the infection caused by *S. flexneri* by inhibiting the expression of N-WASP, which in turn restricts bacterial actin-based motility, stops cell-to-cell spread, and attenuates intracellular infection [13]. In contrast, the expression of miR-29b-2-5p promotes the production of filopodia in host cells by targeting Unc-5 Netrin Receptor C (UNC5C), which enhances bacterial uptake [76]. 

Despite the similarities between *Shigella flexneri* and *Salmonella enterica* serovar Typhimurium, the control of the expression of specific miRNAs elicited by both pathogens completely differs [13]. In particular, miR-let-7i-3p targets the host RGS2 protein and modulates vacuolar trafficking during *S.* Typhimurium infection, inhibiting its pathogenesis [13]. 

In addition, the miR-15 family of miRNAs is very important in the pathogenesis of *S.* Typhimurium, as they are downregulated during specific stages of the infection to allow bacterial spreading [54]. In particular, the miR-15 family arrests the cell cycle of infected cells through the inhibition of the transcription factor E2F1 and derepression of cyclin D1 [54,77]. 

#### 2.2.6. *Burkholderia pseudomallei*

It is also important to consider that the balance between pro-infection and anti-infection miRNAs may be determinant in the fate of intracellular bacterial pathogens. For example, *Burkholderia pseudomallei* downregulates the expression of miR-30b/30c, which results in the upregulation of Rab32. This GTPase promotes the fusion between phagosomes and lysosomes by releasing hydrolases that limit the intracellular growth of *B. pseudomallei* [30]. However, the expression of miR-3473 is triggered during *B. pseudomallei* infection of macrophages, which is mediated by the overexpression of TNF receptor-associated factor 3 (TRAF3) and subsequent TNF-α release, favouring bacterial replication [26]. 

#### 2.2.7. *Listeria monocytogenes*

Listeriosis triggers the upregulation of miR-146a, and the silencing of this miRNA reduces the pathogen’s ability to colonise macrophages intracellularly [78]. At the same time, miR-21 is also activated during infection and controls the polarisation of macrophages [79], which eventually leads to a reduction in the intracellular survival of *L. monocytogenes* [37]. In contrast, miR-26a controls the infection of *L. monocytogenes* by targeting the Ephrin receptor tyrosine kinase 2 (EphA2) to inhibit the internalization or the phagosomal escape of the pathogen [38]. Intriguingly, EphA2 is also an invasion receptor for *C. trachomatis* or *S. aureus* [80,81].

## 3. Novel Antimicrobial Treatments Based on miRNA-Based Technology

RNA-based technology is becoming a feasible strategy to control bacterial infections [82,83,84,85,86]. This new approach has been successfully tested against pulmonary tuberculosis by employing siRNAs targeting *tfgb1* [87]. However, there are now many other opportunities to develop antimicrobial strategies based on other small RNAs, such as many of the miRNAs listed in the previous section. 

In addition, the expression of anti-miRNAs targeting specific miRNAs that facilitate bacterial infection may delay or disrupt the pathogen’s host colonisation. Anti-miRNAs are artificially produced single-stranded RNAs that are complementary to target miRNAs and block their functioning [82]. This strategy has been previously applied in the context of viral infections [88,89]. For example, miravirsen is an anti-miRNA that targets miR-122, an essential miRNA during hepatitis C virus (HCV) infection. Results from a phase II clinical trial indicate that miravirsen can reduce the viral load in a dose-dependent manner [90,91]. The same strategy could be potentially applied to silence miRNAs that are essential for the replication of bacterial intracellular pathogens. 

However, there are some important challenges in the clinical application of miRNA as anti-infectives. The most significant handicap of miRNA therapies is their off-target effects. This could be due to miRNA interactions in a non-specific manner with partially complementary mRNAs [10], leading to important side effects in the host [84]. 

Moreover, the delivery of miRNAs to infected cells could be complicated by the presence of RNAses that can quickly degrade them. This could be partially solved by improving the delivery method of microRNAs to reach specific targets at the cellular or even subcellular levels. This problem has been approached from different perspectives, including the use of nanoparticles, viral delivery systems, high-density lipoproteins, liposomes, or exosomes [84,85], which can facilitate their delivery to host cells [92,93].

Currently, lipid nanoparticles are the leading non-viral delivery systems in the clinical setting [94]. Liposomes are a group of lipid particles that are extensively used to guide RNA-based therapies [95]. However, the main disadvantage of liposomes is the difficulty in functionalising their lipid bilayer [96]. Thus, naturally produced extracellular vesicles are now considered an exciting alternative to improve miRNA delivery (Figure 1).

In particular, exosomes are an up-and-coming solution since they are not toxic and have low antigenicity because they are part of the natural intercellular communication pathways [97]. Exosomes are part of the vesicles generated within the endosomal system and then secreted to the extracellular milieu with essential roles in cell-to-cell communication. Exosomes may efficiently protect the miRNA molecules from degradation by nucleases. Because of this, their use for the delivery of treatments based on nucleic acids is rapidly increasing [98]. In addition, exosomes have advantages over other delivery strategies, such as those based on adenoviruses that may be neutralised by antibodies [97]. 

Similar to other RNA-delivery systems, exosomes must be modified to target infected cells [31,75]. Cancer research has provided different molecular strategies to increase the specificity of exosomal RNA delivery [99]. The main interactions between exosomes and target cells are mainly based on tetraspanins, integrins, lipids, lectins, heparan sulphate proteoglycans, and extracellular matrix elements [98]. 

Interestingly, the isolation of exosomes naturally produced by specific cells increased their fusion with the same parental cells. Thus, isolating exosomes derived from tumour cells and loaded with anti-cancer drugs resulted in well-targeted drug delivery [99]. Moreover, changes in the transmembrane proteins present on the surface of exosomes result in a better adhesion to targeting cells [100]. In addition, the rationale design of exosomes with different membrane modifications also showed promising results in vitro and in vivo in cancer therapies [101]. The use of carbonate apatite or glycan polymers has improved the target cell selectivity by increasing the delivery from endosomes to the cytosol of target cells. Thus, the use of carbonate apatite increased the delivery into the liver, and poly-L-lysine-lactose increased the uptake for hepatocytes [102]. 

Similar approaches could be used to target bacterial-infected cells. However, the development of exosomes as an efficient RNA-delivery system to treat bacterial infections is still in its early stages [103]. During bacterial infection, both the pathogen and eukaryotic cells can produce exosomes that stimulate the immune system or facilitate bacterial infection [104,105]. In addition, exosomes derived from cells primed with bacterial lipopolysaccharide (LPS) could target specific macrophage populations more efficiently and elicit their activation [106]. This strategy may increase the specificity of exosomal-delivery of small RNAs and lower the minimal inhibitory concentration of exosomes required to block host cell infection caused by intracellular pathogens [107,108,109]. Nonetheless, more research is needed to develop an efficient, scalable, easy to produce, stable and specific small RNA delivery system that could be used in the context of bacterial infection. 

## 4. Conclusions

Bacterial and viral infections cause millions of deaths, worldwide, each year. Moreover, the increasing incidence of antimicrobial multidrug-resistant bacteria urgently requires the development of alternative therapeutic strategies to classical antibiotherapy. Here, we reviewed the importance of miRNAs during bacterial infections and new potential strategies to control diseases caused by these pathogens based on these small RNA molecules. Altogether, the data reviewed here highlight the complexity of the interactions between host miRNAs and bacterial pathogens and give a realistic perspective on the scientific community’s insufficient knowledge about miRNAs as host-directed therapies. However, this information also highlights the importance of some miRNAs in host cell immune and antibacterial responses, which could be targeted for developing new antimicrobial therapies. Despite being a very new technology that has hardly been used in the clinic, miRNAs and artificial anti-miRNAs may have a promising future in human medicine.

## Figures and Tables

**Figure 1 antibiotics-11-00356-f001:**
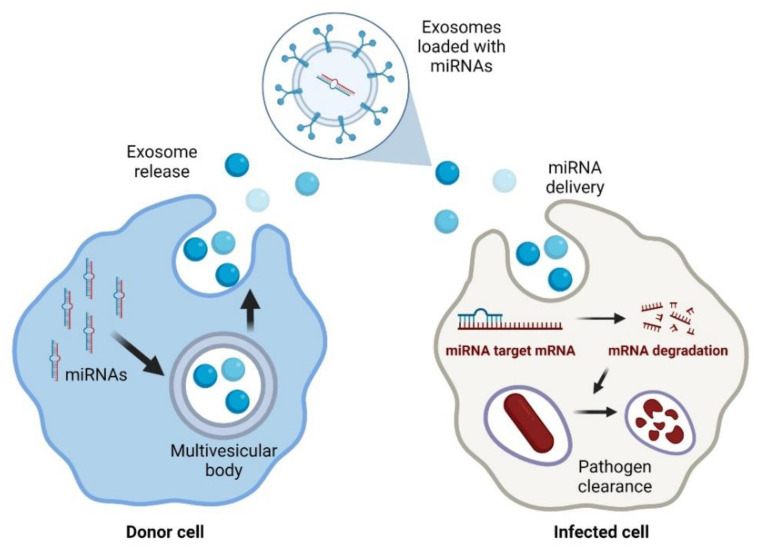
Exosomal delivery of antimicrobial miRNAs to infected cells. Created with BioRender.com (accessed on 1 February 2022).

**Table 1 antibiotics-11-00356-t001:** List of microRNAs identified during infection and their mechanisms of action.

Pathogen	miRNA	Targets	Mechanism of Action	References
Adherent–Invasive *E. coli*	↑ miR-30c and miR-130a *	↓ ATG5 and ATG16L1 *	Inhibits autophagy, facilitates bacterial intracellular survival	[29]
*Burkholderia pseudomallei*	↑ miR-30b/c	↓ Rab32	Stops phagosome maturation, facilitates bacterial intracellular survival	[30]
	↑ miR-3473	↓ TRAF3	Activates TNF-α release, cell apoptosis and inflammatory response, facilitates infection	[26]
*Chlamydia trachomatis*	↑ miR-30c-5p	↓ Drp1	Inhibition of mitochondrial fission to maintain ATP production, facilitates intracellular survival	[31]
	↑miR-9, miR-19 and miR-451	↑ NF-κB pathway	Inflammation control	[32]
	↑ miR-155 and ↓ miR184	↓ Wnt pathway	Inflammation control	[33]
*Francisella tularensis*	↑ miR-155	↓ MyD88 and SHIP-1	Downregulates the TLR adapter protein MyD88 and the inositol 5′-phosphatase SHIP-1 to inhibit the inflammatory response during infection	[34]
*Helicobacter pylori*	↑ miR-25 ↑ miR-155	↓ KLF2 ↓ MyD88	Kruppel-like factor 2 (KLF2) is a direct target of exosome-transmitted miR-25 in vascular endothelial cells, which may contribute to chronic heart disease Reduction of pro-inflammatory cytokine IL-8	[35]
*Legionella pneumophila*	↑ miR-125b, miR-221, and miR-579	↓ DDX58, TP53, LGALS8 and MX1	Three miRNAs govern expression of the cytosolic RNA receptor DDX58, the tumor suppressor TP53, the antibacterial effector LGALS8, and the antiviral factor MX1	[36]
*Listeria monocytogenes*	↑ miR-21	↓ MARCKS and RhoB	The pro-phagocytic regulators myristoylated alanine-rich C-kinase substrate (MARCKS) and Ras homolog gene family, member B (RhoB) are downregulated to hinder pathogen internalization	[37]
	↑ miR-26a	↓ EPHA2	The downregulation of EPHA2 attenuates intracellular survival	[38]
	↑ miR-29	↓ IFN-γ	Suppresses the immune response by downregulating the expression of interferon-γ	[39]
*Mycobacterium bovis* (BCG)	↑ miR-144-3p	↓ ATG4a	Inhibition of autophagy, facilitates intracellular survival	[40]
*Mycobacterium tuberculosis*	↓ miR-17-5p	↑ Mcl-1 and ↑ STAT3	Autophagy activation increasing the interaction of Mcl-1 and Beclin-1	[41]
	↑ miR-18a	↓ ATM	Inhibition of autophagy, facilitates intracellular survival	[42]
	↑ miR-20a-3p	↓ IKKβ	Suppression of immune response, facilitates intracellular survival	[43]
	↓ miR-20b-5p	↑ Mcl-1	Inhibits apoptosis, facilitates intracellular survival	[44]
	↑ miR-27	↓ CACNA2D3	Autophagy inhibition by means of Calcium associated transporters	[45]
	↑ miR-33	↓ ABCA1, CROT, CPT1, HADHB and PRKAA1	Inhibiting cellular cholesterol transport and fatty acid oxidation	[46]
	↑ miR-99b	↓ Inflammatory cytokines	Inhibition of inflammation via MyD88 signaling	[47]
	↓ miR-147 and miR-148a	↑ Inflammatory cytokines	Inflammasome activation	[48,49]
	↑ ↓ miR-155 ^#^	↑ SHIP1/Akt Pathway ↓ Rheb	Cytokine activation and control of autophagic flux	[50,51]
	↑ miR-1178	↓TLR4-pathway	Blocks immune response	[52]
	↑ miR-1958	↓ Atg5	Reduction of autophagy	[53]
*Salmonella* Typhimurium	↑ miR-let-7i-3p	↓ RGS2	Inhibits bacterial replication by the modulation of endolysosomal trafficking and the vacuolar environment	[13]
	↓ miR-15	↓ E2F1 ↑ Cyclin D1	Control of cell cycle progression, which facilitates host cell infection	[54]
	↑ miR-29a	↓ CAV2	Caveolin 2 downregulation results in increased bacterial uptake	[55]
*Shigella flexneri*	↑ miR-29b-2-5p	↓ UNC5C	Enhances filopodia production, facilitating bacterial capture and uptake	[13]
	↑ miR-3668, miR-4732-5p and miR-6073	↓ NWASP	Impairs bacterial actin-based motility, stops cell-to-cell spread, attenuates intracellular infection	[13]
*Staphylococcus aureus*	↑ miR-127	↑ STAT3 ubiquitination	Interleukin activation and bacterial clearance	[56]
*Vibrio cholerae*	↑ miR-155 and miR-146a	↓ NF-κB pathway	Reduction of inflammatory and immune responses in intestinal epithelial cells	[57]
Broad-spectrum miRNas	↑ miR-29 ↑ miR-124 ↑ miR-302b	↓ IFN-γ ↓ TLRs/NF-κB ↑ Cytokine genes	Inhibition of the immune response Inhibition of the immune response Activates the immune response	[39] [25,58,59] [60]
Lipopolysaccharide	↑ miR-155 and miR-146a	↓ TLR4 pathway	Negative-feedback loop of the TLR4-mediated cellular response in human monocytes exposed to lipopolysaccharide (LPS)	[28]

* The symbol ↑ represents upregulation during infection, whereas ↓ means downregulation. ^#^ The role of mir-155 in tuberculosis is host cell-specific.

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
