# Peer review of "Understanding microRNAs in the Context of Infection to Find New Treatments against Human Bacterial Pathogens"

_antibiotics, 2022, doi:10.3390/antibiotics11030356_

Round 1

Reviewer 1 Report

The authors Mourenza et al in their review titled "Understanding microRNAs in the context of infection to find new treatments against human bacterial pathogens" have focused on reviewing the role of miRNAs against bacterial pathogens. The review covers aspects but in order for completeness would need to actually subdivide the review into various subheadings and also explain in detail certain aspects that are mentioned below for completeness:

  1. The introduction is quite incomplete. More information would be needed on history of small RNAs and how they were considered to be potential targets for the development of antibacterials. A more comprehensive explanation or historical account is needed so as to obtain interest in the role of miRNAs as antibacterials.
  2. miRNAs vs disease types would be suitable headings under which the role of specific mRNAs against bacterial pathogens needs to be elaborated an not as one large paragraph as detailed in the "Human miRNAs and pathogen infections" subparagraph. Group based on Tuberculos, or Tularensis etc. Additional references are also needed to justify the different approaches against the pathogens.
  3. Modulation of the immune environment of mode of action is completely missing and the authors would need to get that as a separate section. The role of the immune environement is one of major importance that regulates the action against pathogens. Though some mention is made in the text, a detailed explanation is needed for the same to be of benefit to the reader. Or if not immune modulation then detailed explanation of mode of action would be necessary.
  4. Images on mode of action would be essential.
  5. Images/tables on classes of molecules and pathogen if sufficient data can be provided would be of help.
  6. Tabulation of novel treatment options against pathogens also needs to be expanded and tabulation of the elements would be essential for the completeness of the review.

Author Response

Reponses to Reviewer 1

The authors Mourenza et al in their review titled "Understanding microRNAs in the context of infection to find new treatments against human bacterial pathogens" have focused on reviewing the role of miRNAs against bacterial pathogens. The review covers aspects but in order for completeness would need to actually subdivide the review into various subheadings and also explain in detail certain aspects that are mentioned below for completeness:

1. The introduction is quite incomplete. More information would be needed on history of small RNAs and how they were considered to be potential targets for the development of antibacterials. A more comprehensive explanation or historical account is needed so as to obtain interest in the role of miRNAs as antibacterials.

Many thanks for this comment, we have expanded the introduction accordingly, following the reviewer indications (lines 65-91). This has indeed created a more comprehensive introduction and we have put in context the rest of the information included in the review. We are therefore thankful to the reviewer for the excellent suggestion.

2. miRNAs vs disease types would be suitable headings under which the role of specific mRNAs against bacterial pathogens needs to be elaborated an not as one large paragraph as detailed in the "Human miRNAs and pathogen infections" subparagraph. Group based on Tuberculos, or Tularensis etc. Additional references are also needed to justify the different approaches against the pathogens.

We agree with the reviewer. The initial version was difficult to follow. We have divided the section into two subsections in response to point 3. In addition, each of these subsections has now subheadings to divide the text per pathogen, which has greatly enhanced the narrative structure of the revision. Again, this has improved massively our review, thanks.

3. Modulation of the immune environment of mode of action is completely missing and the authors would need to get that as a separate section. The role of the immune environement is one of major importance that regulates the action against pathogens. Though some mention is made in the text, a detailed explanation is needed for the same to be of benefit to the reader. Or if not immune modulation then detailed explanation of mode of action would be necessary.

In agreement with this comment, we have divided section 2 into two subsections. Section 2.1 is focused on the immune response against bacterial pathogens, mainly related to inflammation. Whereas section 2.2 is focused on the effect of the miRNA on the cellular pathways governed by intracellular bacterial pathogens during host cell infection. This has clearly separated these two aspects of the role of miRNAs on bacterial infection and the modification is very beneficial to the reader, thanks.

4. Images on mode of action would be essential.

We have covered many different modes of action of different miRNAs and in the context of very distinct infections. Therefore, this request would require a very complex figure that may not be very informative to the reader. However, we have introduced this information in Table 1, which has been expanded and it is now including a detailed description of the mechanism of action of miRNAs involved in bacterial infection.

5. Images/tables on classes of molecules and pathogen if sufficient data can be provided would be of help.

As mentioned above, table 1 is including this information in detail.

6. Tabulation of novel treatment options against pathogens also needs to be expanded and tabulation of the elements would be essential for the completeness of the review.

The information on novel treatment options based on miRNAs against bacterial pathogens is very scarce. We have used section 3 to provide a general overview of the therapeutic possibilities that this new technology is opening. We have also included Figure 1 to provide an illustration of the exosomal delivery of miRNAs towards infected cells, which is the most promising option at this stage to produce novel antimicrobial therapies based on miRNA technology.

Reviewer 2 Report

1. There should be one figure, which reflects the whole concept in this manuscript like graphical abstract.

2. There are 145 related articles in the literature to make sure include all important articles during the revision process.

3. Line 131 needs a reference

  1. pylori chronic infection can cause other disorders such as coronary heart disease." 

4. Grammatical mistakes:

Line 123:  "a markedly decrease...."  It should be changed to: 

"a marked decrease...."

Line 147:   "the expression miR-127...."

It should be changed to:

 "the expression of miR-127...."

Author Response

Responses to Reviewer 2

  1. There should be one figure, which reflects the whole concept in this manuscript like graphical abstract.

 Many thanks for this comment, we have added figure 1, which summarizes the concept of exosomal delivery of antimicrobial miRNAs to bacterial infected cells. This has greatly improved the visual impact of the manuscript, and therefore we are thankful to the reviewer for this comment.

  1. There are 145 related articles in the literature to make sure include all important articles during the revision process.

 Thanks for this comment, we have expanded the number of citations to 109 in the current version. However, we have not included miRNAs related to bacterial pathogens causing only animal infections in this manuscript. The main objective of this revision was to update the field in the latest developments on antimicrobial miRNA based technology with promising applications in human medicine.

  1. Line 131 needs a reference
  1. pylori chronic infection can cause other disorders such as coronary heart disease." 

 Thank you, this has been addressed in line 154 by adding reference 44 to the manuscript.

  1. Grammatical mistakes:

Line 123:  "a markedly decrease...."  It should be changed to: 

"a marked decrease...."

 Modified accordingly, thanks (Line 132).

Line 147:   "the expression miR-127...."

It should be changed to:

 "the expression of miR-127...."

Thank you for spotting this mistake, we have now modified this sentence following your suggestion (line 144).

Round 2

Reviewer 1 Report

The authors have addressed all issues